# Effects of Combined High Temperature and Waterlogging Stress at Booting Stage on Root Anatomy of Rice (*Oryza sativa* L.)

**Bo Zhen** [1,2], **Huizhen Li** [1], **Qinglin Niu** [1], **Husen Qiu** [1], **Guangli Tian** [1], **Hongfei Lu** [1] and **Xinguo Zhou** [1,*]

1   Farmland Irrigation Research Institute, Chinese Academy of Agricultural Sciences, Xinxiang 453002, China; zhenbo@caas.cn (B.Z.); lihuizhen@caas.cn (H.L.); niuqinglin@caas.cn (Q.N.); qiuhusen@caas.cn (H.Q.); tianguangli@caas.cn (G.T.); luhongfei@caas.cn (H.L.)
2   National Agro-Ecosystem Observation and Research Station of Shangqiu, Shangqiu 476000, China
*   Correspondence: zhouxinguo@caas.cn

**Abstract:** In recent years, the alternating occurrence of high temperature and waterlogging disasters in South China has seriously reduced the yield of single cropping rice. Studying the changes in anatomical structure of the rice root system could provide theoretical basis for understanding the mechanisms of high temperature and waterlogging stress. To examine interactions between temperature and waterlogging stress, an experiment was set up in a growth chamber consisting of two temperatures (moderate, 30–34 °C and high, 35–38 °C) with three depths of flooding (0–5, 10 and 15 cm). Treatments commenced at the booting stage and lasted five days, after which all treatments were returned to a 0–5 cm flooding depth and the same temperature regime. Observations were made immediately after cessation of treatments, then after 5, 10 and 20 d to test the effect of treatments on subsequent root anatomical development. The low-stress control (0–5 cm, medium temperature) showed no change with time in aerenchyma area, thickness of the outer root, stele diameter, and the number nor diameter of xylem vessels. Root diameter and stele diameter under the high-stress control (0–5 cm, high temperature) were decreased by 29.09% and 15.28%, respectively, at the booting stage, whereas the high stress control (15 cm, high temperature) affected only the vessel diameter, reducing it by 14.11% compared with that in the low-stress control (0–5 cm, medium temperature). Compared to the high-stress control (0–5 cm, high temperature), the interaction of high temperature and waterlogging stress alleviated the inhibiting effect of the changes in the root system, especially after the end of the stress. We thought that waterlogging could reduce the damage of high temperature on rice root growth. Low water depth waterlogging has little effect on rice root system and aerenchyma area root diameter at 0, 5, 10 and 20 d after the stress ended, and the thickness of the outer root, stele diameter and the number and diameter of vessels at 0 d under M15 (15 cm, medium temperature) had no significant difference compared with M5 (0–5 cm, medium temperature). However, the increase in rice root diameter, stele diameter, thickness of the outer root cortex depth and vessel diameter were inhibited under high temperature stress at the booting stage. Root diameter and stele diameter under H5 (0–5 cm, high temperature) were decreased by 29.09% and 15.28%, respectively, at the booting stage, whereas H15 (15 cm, high temperature) affected only the vessel diameter, reducing it by 14.11% compared with that in the M5. Compared to H5, the interaction of high temperature and waterlogging stress alleviated the inhibiting effect of the changes in the root system, especially after the end of the stress. We thought that waterlogging could lighten the damage of high temperature on rice root growth.

**Keywords:** rice; high temperature; waterlogging; root anatomical structure

## 1. Introduction

China is one of the major rice producers, accounting for 18.5% of the world's total planting area and 27.7% of world rice production [1]. Temperature and moisture are the main factors affecting rice production. Due to global warming, the atmospheric temperature is expected to increase by 0.3–0.7 °C by the middle of this century [2]. Climate change impacts the underground processes in ecosystems—increased soil temperature may alter the growth of underground plant parts [3,4]. High temperatures pose a serious threat to global food security. In rice production, a high temperature (>35 °C), especially during grain filling, can remarkably reduce grain development, quantity and quality [5]. High temperature can affect many processes of rice growth and development, including germination, seedling growth, leaf emergence, tillering, heading and filling [6–11]. The frequency of extremely high temperatures and heat waves in most of mainland China is likely to continue increasing in the 21st century [12]. According to Wu Qixia, the middle and lower reaches of the Yangtze River were exposed to a combination of heat and waterlogging stress that occurred 15 times between 1960 and 2010, whereas there were only a further 19 crop failure events that were caused by a single one of these stresses, accounting for 44% [13]. In recent years, high temperature and rainstorms have coincided frequently with the rice reproductive growth stage in Huanghuai and South China [14]. After heavy summer rainfalls, the water cannot be completely drained in a short period of time and instead of the shallow flooding (<5 cm) for which the rice plant is anatomically adapted, the surplus water causes deeper ponding, resulting in yield loss. As these heavy rains are typically followed by high temperatures, rice is thus subjected to the dual stresses of high temperature and waterlogging, which affect the absorption and utilization of nutrients by rice roots and lead to reduction in rice yield. Root growth is a dynamic process and root architecture may change in response to alterations in the immediate environment [15]. Therefore, a study on the combined effect of high temperature and waterlogging stresses on the microstructure of the rice root system will help to elucidate the growth and development of the root system under these dual stresses from an anatomical perspective. This knowledge will provide theoretical reference for improving rice yield stability and stress resistance and ensure future food security [16].

In a drought or flooded environment, plants can adapt to adversity by changing homeostatic mechanisms and root architecture [17]. Shallow flooding increases root length, root surface area, root volume and root bifurcation rate, but the average root diameter decreases. Roots are most susceptible to suffer from oxygen shortage under waterlogging and flooding conditions [18]. Previous studies have shown that five days of flooding at the tillering stage triggers the formation of aeration tissues in rice roots and the development of thick-walled cells in the root cortex. At the same time, flooding also inhibits the development of the rice outer root cortex depth [19], thus affecting the absorption and transportation of water and nutrients by rice roots. Root morphological and anatomical adjustments have been revealed to be important to improve oxygen supply to maintain root function under oxygen-deficient conditions.

High air temperatures lead to high soil temperatures, which in turn affect root water absorption. High root temperature accelerates the aging process, causing the lignified section of the root to extend almost to the tip, leading to a decrease in the absorption area of the root and the rate at which the root can absorb water and nutrients. According to Kang shaozhong [20], the total water cross-section of roots in a high temperature environment was smaller than that in a low temperature environment during the jointing stage of maize, so the root hydraulic conductivity was reduced.

Substantial research has been undertaken on the effect of a single stress on the water absorption capacity of rice roots, especially on changes in rice root anatomical structure. Anatomical features of the rice root system include root diameter, cross-section of aeration tissue, stele diameter, thickness of thick-walled cells in outer cortex and cross-sectional area of cells, all of which affect the absorption and transportation of water and nutrients in the root system [21]. Both high temperature and deep flooding have a negative effect on rice. When rice encounters high temperature, water consumption and soil temperature increase significantly. If the water depth is kept high (more than 5 cm), the

soil temperature can be reduced, and the water absorption of rice roots can be ensured. Previous studies have shown that during rice growth, daily average temperatures higher than 32 °C, or daily maximum temperatures higher than 35 °C, lead to heat damage of rice [22], whereas five days of flooding and more than 10 cm of flooding depth affect the growth of rice and the microstructure of its root system [19], which showed that the double stress of high temperature and waterlogging exists in reality. If we ignore the interaction, and focus on the two stresses separately, their negative effects are magnified. For some dryland crops, the dual stresses of high temperature and waterlogging intensify the damage to crops, but for wetland crops, the interaction between them is not clear. We hypothesized that high temperature and waterlogging may play a positive role in the root growth of rice.

In this study, we analyzed the effects of high temperature and waterlogging on the growth and development of rice roots by comparing the root diameter, stele diameter and the number and diameter of vessels between plants exposed to different temperatures and waterlogging regimes. We hope to expand the understanding of interaction between high temperature and waterlogging stresses on rice, and to find some evidence that is conducive to the development of the rice root.

## 2. Materials and Methods

### 2.1. Plant Material and Test Site

The experiment was carried out at the National Field Science Observatory of Shangqiu Ecosystem in Henan Province from May to October 2017. The average annual rainfall, annual average evaporation and daily maximum temperature at the station are 705.1 mm, 1751 mm and 32 °C, respectively. The plant material tested was the middle–late maturing conventional japonica rice variety 008, whose growth period lasts about 145 days along the Huaihe River. In this experiment, rice plants were planted in barrels. The bottom diameter of the barrels was 21.5 cm, the upper diameter was 25 cm and their depth was 29.5 cm. Test soil of the fluvo-aquic type was collected from the field (0–20 cm soil layer) at the research station. After air-drying and sieving, the soil was loaded into barrels at a volume of 10 kg air-dried soil per barrel. The amount of fertilizer applied per barrel was: 2.20 g $CO(NH_2)_2$, 0.90 g $K_2SO_4$, 2.50 g $KH_2PO_4$ and 16.7 g organic fertilizer.

### 2.2. Experimental Design

Seeds were sown on 4 May 2017. After 40 days (13 June), seedlings with identical growth were selected for transplantation. Three holes were made in each barrel, and two plants were transplanted in each hole. At the tillering stage and jointing stage, rice grew under a rain-proof shelter and maintained a water depth of 0–5 cm (from 13 June to 4 August). Experimental treatments of high temperature and flooding depth were imposed at the booting stage of rice, beginning at 6:00 on 4 August, then all treatments were moved to a rain-proof shelter at 18:00 on 9 August, maintained at water depth of 0–5 cm and harvested on 24 October 2017. Except for temperature and waterlogging regimes, other agrotechnical measures were the same in each treatment.

Considering the meteorological conditions in the Huaihe River area from July to September (Figure 1), the experiment adopted a completely random design with two factors, high temperature and flooding. Temperature treatments were moderate (30–34 °C) and high (35–38 °C), while flooding treatments were 0–5, 10 and 15 cm. Treatments were designated M5, M10 and M15, respectively, for the moderate temperature and H5, H10 and H15 for the high temperature. Temperature treatments were imposed in a climate chamber. The set time of the high temperature treatment was 10:00–18:00. The temperature range was 35–38 °C (Figure 2). Between 10:00–13:00, the temperature increased by 1 °C per hour, and then decreased by 1 °C per hour from 14:00 to 18:00. At the other time, the treatment temperature was the same as M5. Relative humidity was set to 80%, the illumination period was from 06:00 to 19:00 and illumination intensity was 1000 µmol/m²/s. Flooding stress was simulated in a plastic water tank (970 × 770 × 670 mm). Each treatment was repeated 20 times (20 barrels), for a total of 120 barrels.

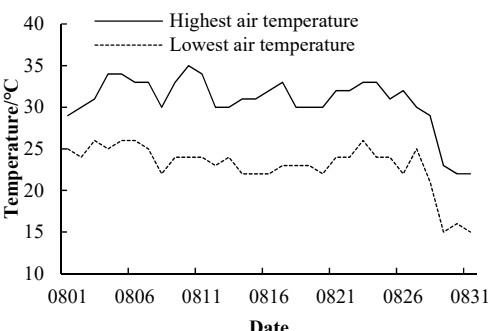

**Figure 1.** Daily maximum and minimum temperature in August in 2017.

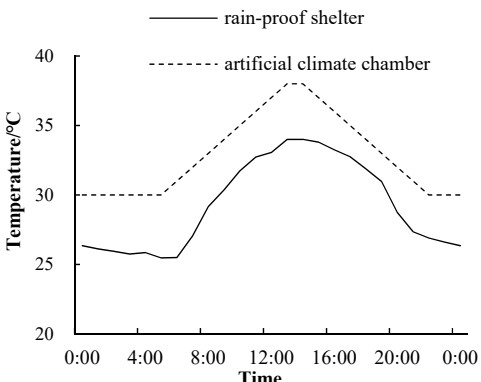

**Figure 2.** Daily variation of air temperature.

*2.3. Measurement of Different Root Anatomical Parameters and Their Analysis*

2.3.1. Root Anatomical Observations

Three pots of rice plants with identical growth were selected for each treatment at 0, 5, 10 and 20 days after the stress ended. Three new, white roots were selected from each pot, and 5–10 mm segments of each root was excised 10–15 mm away from the root tip and dropped immediately into FAA (Formal dehyde-acetic acid-ethanol) fixative (40% formalin:glacial acetic acid:70% ethanol, 1:1:18, by vol). After dehydration with a graded alcohol series and clearing with xylene, the material was immersed in wax, and the wax-soaked tissues were embedded in the embedding machine. Two wax blocks, each comprising three samples, were prepared for each treatment. The prepared wax blocks were cut into 3 μm thick slices using a microtome. Paraffin sections were then placed in saffron stain for 1–2 h and slightly washed with tap water to remove the excess dye. For discoloration, samples were sequentially placed into 50, 70 and 80% gradient alcohols for 3–8 s each. Slices were placed into solid green dyeing solution for 30–60 s and dehydrated (Dehydrator, Wuhan Junjie Electronics Co., Ltd., JJ-12J, Wuhan, China) in anhydrous ethanol. The sections were stained with safranin fixed green dye and mounted with neutral gum. Slices were observed with a positive optical microscopy (NIKON ECLIPSE E100, Nikon, Tokyo, Japan) and imaging system (NIKON DS-U3, Nikon, Tokyo, Japan). Relevant indexes of the root system were measured with Case Viewer 2.0 software (3DHISTECH Ltd., Budapest, Hungary), and clear pictures were selected for photography.

2.3.2. Data Statistical Methods

Root diameter (RD, μm): six cross-sectional areas of root cells were measured by CaseViewer 2.0 (3DHISTECH Ltd., Budapest, Hungary) and three measurements of similar value were selected to



calculate cross-sectional root diameter using Equation (1). The average value of the cross-sectional area was labeled for other parameters.

$$\text{RD} = 2 \times \sqrt{\frac{S_1}{3.14}} \tag{1}$$

where RD is the root diameter, μm and $S_1$ is the cell cross section area.

Thickness of the outer root (μm): generally, epidermis, outer cortex, thick-walled cells and adjacent cortical parenchyma cells are collectively referred to as the outer cells of the roots [23]. Thickness of the outer root at 15 mm from the root tip reflects the development of rice roots. Using the measuring tool in CaseViewer 2.0 (3DHISTECH Ltd., Budapest, Hungary), three labeled root cells (root cells with the same cross-sectional diameter) were selected. Four values in the direction of 0:00, 3:00, 6:00, and 9:00 were selected for each root cell volume. The thickness of the outer cells was measured in 12 roots, and an average value was calculated.

Stele diameter (SD, μm): root stele cells refer to the stele parts within the endodermis. The root stele area of three labeled root cells was measured by using the measuring tool in CaseViewer 2.0 and the stele diameter was calculated by using Equation (2), the obtained values were then averaged.

$$\text{SD} = 2 \times \sqrt{\frac{S_2}{3.14}} \tag{2}$$

where SD is the Stele diameter, μm; $S_2$ is stele area.

Number and diameter of vessels: when calculating the stele diameter, the number of vessels (VN) and the area of the vessels were recorded and vessel diameter (VD, μm) was calculated by using the Equation (3). The obtained values were then averaged.

$$\text{VD} = 2 \times \sqrt{\frac{S_3}{3.14}} \tag{3}$$

where VD is the vessel diameter, μm and $S_3$ is vessel area.

### 2.3.3. Data Processing and Analysis

Microsoft Excel and SPSS 19.0 software (IBM Corp., Armonk, NY, USA) were used to analyze the data, and significant differences between the means were determined with Duncan's new multiple range test.

## 3. Results

### 3.1. Effects of High Temperature and Waterlogging on Anatomical Structure Parameters of Rice Roots

The effects of high temperature and waterlogging stress on rice root structure had an obvious after-effect (Table 1). 0 day after stress ended (9 August), temperature, water and temperature × water affected significantly the aerenchyma area and vessel diameter ($p < 0.05$). Five days after stress ended (14 August), temperature, water and temperature × water affected significantly the aerenchyma area and vessel diameter. As stress ended for 10 days (19 August), temperature × water affected significantly the anatomical parameters of rice roots. After 20 days (29 August), temperature × water had significant effects on aerenchyma area, thickness of the outer root, stele diameter and vessel diameter.

**Table 1.** *p* value of experimental factors and their interactions on anatomical parameters of rice roots.

| Days after Stress Ended | Experimental Factor | Aerenchyma Area | Root Diameter | Thickness of Outer Root | Stele Diameter | Vessel Diameter |
|---|---|---|---|---|---|---|
| 0 day | T | 0.000 * | 0.000 * | 0.011 * | 0.010 * | 0.000 * |
| | W | 0.000 * | 0.065 | 0.102 | 0.003 * | 0.006 * |
| | T × W | 0.000 * | 0.378 | 0.238 | 0.333 | 0.001 * |
| 5 days | T | 0.000 * | 0.165 | 0.096 | 0.559 | 0.016 * |
| | W | 0.000 * | 0.156 | 0.000 * | 0.032 * | 0.000 * |
| | T × W | 0.000 * | 0.052 | 0.000 * | 0.083 | 0.001 * |
| 10 days | T | 0.042 | 0.255 | 0.237 | 0.141 | 0.043 * |
| | W | 0.000 * | 0.128 | 0.007 * | 0.085 | 0.744 |
| | T × W | 0.000 * | 0.026 * | 0.000 * | 0.024 * | 0.000 * |
| 20 days | T | 0.002 * | 0.001 * | 0.000 * | 0.965 | 0.000 * |
| | W | 0.003 * | 0.047 * | 0.000 * | 0.000 * | 0.012 * |
| | T × W | 0.002 * | 0.427 | 0.000 * | 0.000 * | 0.000 * |

Note: T and W represent temperature and water, respectively; * means significant difference at 0.05 level.

### 3.2. Aerenchyma Formation in Plants Exposed to High Temperature and Waterlogging Stress

#### 3.2.1. Zero Day after the End of Stress

Waterlogging at the booting stage significantly promoted the formation of aerenchyma in rice roots (Figure 3). Compared to the low-stress control (M5), aerenchyma was especially well-developed in M10, H5 and H15 treatments, and the number and volume of aerenchymatous cavities were large and connected. The aerenchyma area was followed as: M10 > H5 > H15, and M 10 was 1.97 and 4.88 times of that of H5 and H15, respectively (Table 2).

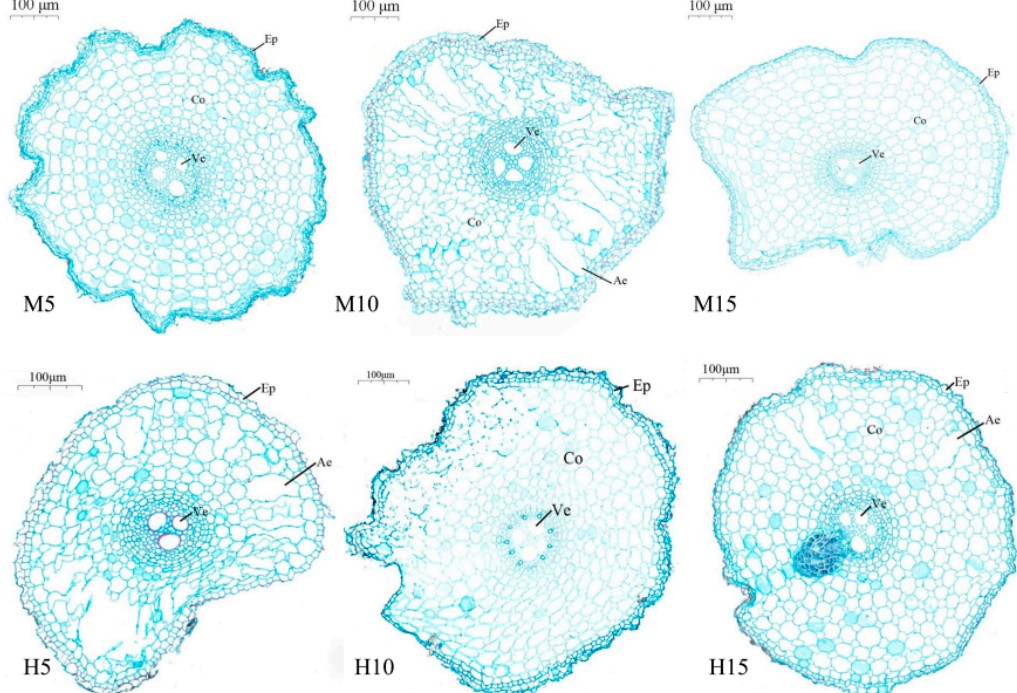

**Figure 3.** Cross-sectional map of root anatomy of rice at 0 day (9 August) after high temperature and waterlogging stress. Note: Ep: epidermis; Co: cortex; Ae: aerenchyma; Ve: vessel.

**Table 2.** Aerenchyma area after exposure to high temperature and waterlogging stress ($\mu m^2$).

| Treatment | 0 Day after the End of Stress | 5 Days after the End of Stress | 10 Days after the End of Stress | 20 Days after the End of Stress |
|---|---|---|---|---|
| M5 | 0.00d | 0.00d | 318,980.13b | 262,343.33a |
| M10 | 62,725.87a | 76,833.10c | 442,274.67a | 242,150.33a |
| M15 | 0.00d | 0.00d | 223,713.24b | 308,096.25a |
| H5 | 31,830.90b | 35,345.34d | 306,063.57b | 300,848.94a |
| H10 | 0.00d | 191,419.50b | 274,994.44b | 147,754.71b |
| H15 | 12,859.79c | 317,942.06a | 315,199.43b | 172,162.13b |

Note: Data are shown as the mean of triplicate measurements. Different letters follow after standard deviation to express significant differences ($p < 0.05$). M5, regular irrigation, moderate temperature (30–34 °C), water depth maintained at 5 cm; M10, moderate temperature, light waterlogging, and M15, moderate temperature, heavy waterlogging. H5, high temperature (35–38 °C), water depth maintained at 5 cm; H10, high temperature × light waterlogging; H15, high temperature × heavy waterlogging. All treatments were conducted at the booting stage. Light waterlogging indicates water depth maintained at 10 cm; heavy waterlogging indicates water depth maintained at 15 cm.

### 3.2.2. Five Days after the End of Stress

At the same flooding depth, high temperature significantly increased the formation of aerenchyma in rice roots, and the dual stresses of high temperature and waterlogging played a positive role in the formation of aerenchyma in rice roots. Five days after the end of stress, compared to the low-stress control (M5), aerenchyma was formed in roots of plants treated with H5, H10, H15, and M10, under high temperature stress and the area of aerenchyma increased with the increase of flooding depth (Figure 4). The aerenchyma area of the H15 treatment was the largest. Under the same flooding depth, the aerenchyma area under high temperature stress was higher than that under moderate temperature. The aerenchyma area of H10 was 2.49 times that of M10 (Table 2).

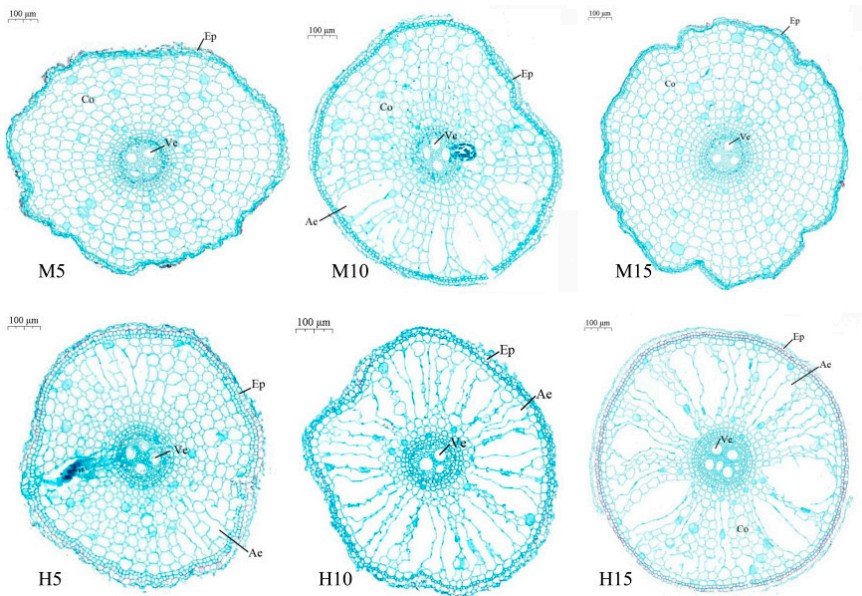

**Figure 4.** Cross-sectional map of root anatomy of rice at five days (14 August) after high temperature and waterlogging stress.

### 3.2.3. Ten Days after the End of Stress

The aerenchyma of all treated roots that was well developed due to the aging of the roots (Figure 5), light flooding (M10) increased the formation of aerenchyma compared to M5. The aerenchyma area of M10 was increased by 38.65% (Table 2).

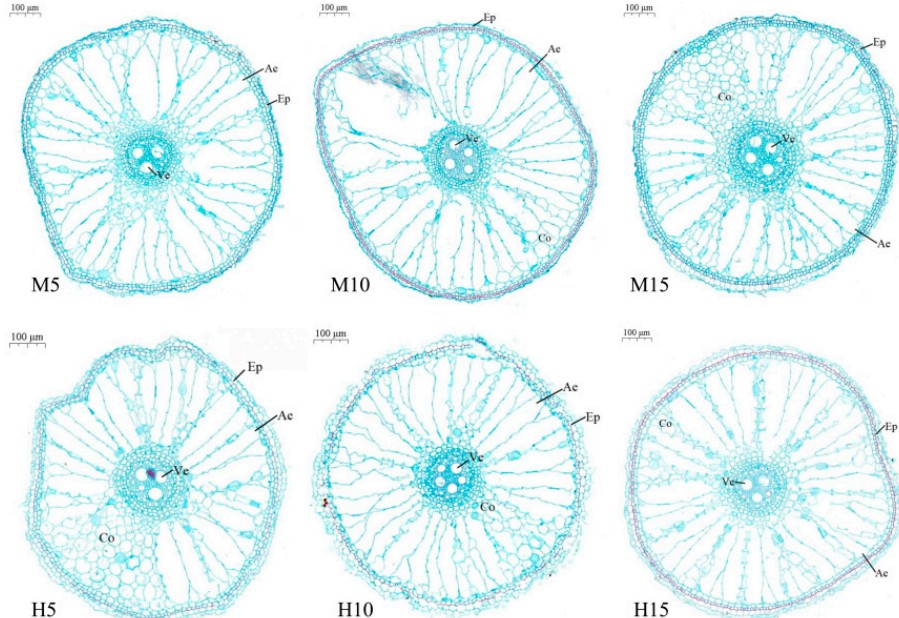

**Figure 5.** Cross-sectional map of root anatomy of rice at 10 days (19 August) after high temperature and waterlogging stress.

### 3.2.4. Twenty Days after the End of Stress

With the advance of the growth period, as the root system aged, twenty days after the end of stress, the dual stresses of high temperature and waterlogging had a negative effect on the formation of aerenchyma in rice roots (Figure 6). Compared to M5, the aerenchyma area of H10 and H15 was decreased by 43.68% and 34.38%, respectively (Table 2).

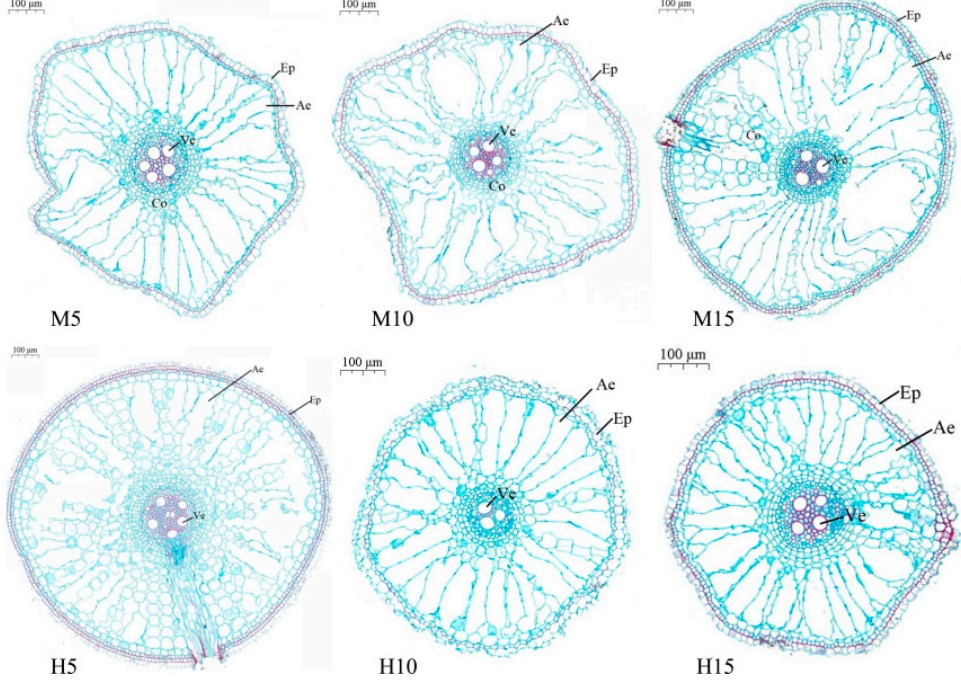

**Figure 6.** Cross-sectional map of root anatomy of rice at 20 days (29 August) after high temperature and waterlogging stress.

### 3.3. Effects of Combined High Temperature and Waterlogging Stresses on Root Diameter

High temperature treatment at the booting stage significantly reduced the root diameter and exhibited certain after-effects (Table 3). Compared to M5, 0 day after the end of stress, the root diameter under the H5 treatment was decreased by 29.09% ($p < 0.05$), Even if the temperature returned to normal for 20 days, the root diameter treated with H5 was decreased by 14.41% ($p < 0.05$).

**Table 3.** Root diameter (μm) after exposure to high temperature and waterlogging stresses.

| Treatment | 0 Day after the End of Stress | 5 Days after the End of Stress | 10 Days after the End of Stress | 20 Days after the End of Stress |
|---|---|---|---|---|
| M5 | 631.83ab | 864.60abc | 888.82b | 788.77ab |
| M10 | 667.91ab | 932.11ab | 993.03a | 766.98bc |
| M15 | 758.19a | 861.05abc | 857.13b | 872.25a |
| H5 | 448c | 732.30c | 834.56b | 675.14c |
| H10 | 570.42bc | 744.79bc | 872.84b | 701.25bc |
| H15 | 539.70bc | 972.12a | 937.45ab | 731.28bc |

Note: Treatments are defined in Table 2. Data are shown as the mean of triplicate measurements. Different letters follow after standard deviation to express significant differences ($p < 0.05$).

We analyzed the relationship between root diameter and soil temperature. We found that when the soil temperature was between 25–35 °C, the root diameter decreased with the increase of soil temperature (Figure 7).

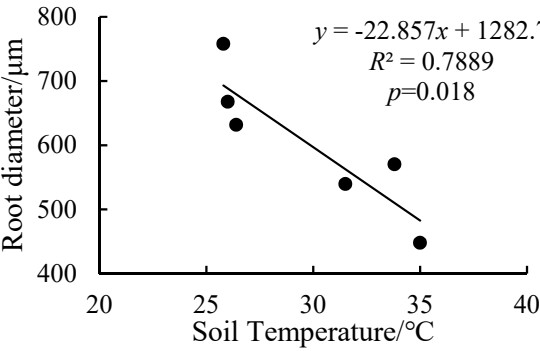

**Figure 7.** Linear regression between soil temperature and root diameter.

### 3.4. Effects of Combined High Temperature and Waterlogging Stresses on Thickness of the Outer Root

The effects of high temperature and waterlogging (H15) on the thickness of the outer layer of the rice heel were most obvious (Table 4). Five days after the end of stress, compared to M5, the thickness of the outer root under the M10 and M15 treatments was increased by 29.69%, and 15.71% respectively, whereas the thickness of the outer root under H5 stress was decreased by 17.78%. However, the thickness of the outer root under the dual stresses of high temperature and heavy waterlogging (H15) was increased by 48.97% compared with M5. Ten days after the end of stress, the thickest thickness of the outer root was observed in H15 (17.87% thicker than that in the M5), and the thinnest thickness of the outer root was detected in H5 (18.38% thinner than that in the M5). Twenty days after the end of stress, the thickness of the outer root treated with H15 was decreased by 10.87% compared with that of M5.

**Table 4.** Thickness of the outer root (μm) after exposure to high temperature and waterlogging stresses.

| Treatment | 0 Day after the End of Stress | 5 Days after the End of Stress | 10 Days after the End of Stress | 20 Days after the End of Stress |
|---|---|---|---|---|
| M5 | 43.2ab | 42.64c | 50.6b | 47.48b |
| M10 | 47.58a | 55.3b | 47.42bc | 45.12bc |
| M15 | 40.44b | 49.34b | 47.54bc | 56.84a |
| H5 | 41.00b | 35.06d | 41.30c | 44.06c |
| H10 | 39.26b | 39.3cd | 51.08b | 41.78c |
| H15 | 37.82b | 63.52a | 59.64a | 42.32c |

Note: Treatments are defined in Table 2. Data are shown as the mean of triplicate measurements. Different letters follow after standard deviation to express significant differences ($p < 0.05$).

### 3.5. Effects of Combined High Temperature and Waterlogging Stresses on Stele Diameter

Submergence was beneficial to increase the diameter of the rice root stele (Table 5). Zero days after the end of stress, compared to M5, the stele diameter treated with the M10 treatment was increased by 19.27%, and the stele diameter under H5 treatment was decreased by 15.28%. Five days after the end of stress, the stele diameter under the H15 treatment was increased significantly by 20.26% compared with M5. Ten days after the end of stress, the stele diameter under the M15 treatment was increased significantly by 15.08% compared with that in the M5 treatment.

**Table 5.** Stele diameter after exposure to high temperature and waterlogging at booting stage in rice (μm).

| Treatment | 0 Day after the End of Stress | 5 Days after the End of Stress | 10 Days after the End of Stress | 20 Days after the End of Stress |
|---|---|---|---|---|
| M5 | 129.98b | 149.03b | 151.43b | 151.73bc |
| M10 | 155.04a | 163.43ab | 155.20b | 144.29bcd |
| M15 | 138.95ab | 157.46b | 174.27a | 167.89b |
| H5 | 110.12c | 147.11b | 156.59b | 205.29a |
| H10 | 135.26b | 153.37b | 154.55b | 121.26d |
| H15 | 134.84b | 179.23a | 153.06b | 138.16cd |

Note: Treatments are defined in Table 2. Data are shown as the mean of triplicate measurements. Different letters follow after standard deviation to express significant differences ($p < 0.05$).

### 3.6. Effects of High Temperature and Waterlogging on Vessel Number and Diameter

The diameters of vessels are easily affected by high temperature and waterlogging (Table 6). Zero days after stress ended, compared to M5, the vessel diameter under the H15 treatment was decreased significantly by 14.11%. Five days after stress ended, root vessel diameter of rice treated with H10 and M10 was increased by 17.36% and 8.57%, respectively, when compared with that of the M5 treatment, while root vessel diameter of rice treated with H15 was lower than that of M5. At 20 days after the end of stress, compared to M5, the vessel diameters under H5, H10 and H15 treatments were decreased by 8.89%, 9.80% and 17.28%, respectively.

**Table 6.** The number and diameter of vessels after exposure to high temperature and waterlogging.

| Treatment | 0 Day after the End of Stress | | 5 Days after the End of Stress | | 10 Days after the End of Stress | | 20 Days after the End of Stress | |
|---|---|---|---|---|---|---|---|---|
| | VN | VD/μm | VN | VD/μm | VN | VD/μm | VN | VD/μm |
| M5 | 3.67a | 29.50ab | 4.00ab | 34.44cd | 3.00b | 37.06a | 4.33a | 35.31a |
| M10 | 3.67a | 31.02a | 4.00ab | 37.39b | 4.33a | 36.67a | 5.00a | 30.70bc |
| M15 | 3.67a | 31.37a | 3.33b | 33.48de | 5.00a | 34.00c | 4.00a | 35.09a |
| H5 | 3.33a | 28.48b | 4.00ab | 35.18c | 4.33a | 33.53c | 4.67a | 32.17b |
| H10 | 4.00a | 30.21ab | 3.33b | 40.42a | 4.00a | 34.57bc | 3.00b | 31.85b |
| H15 | 3.67a | 25.34c | 5.00a | 32.65e | 4.00a | 36.25ab | 4.00a | 29.21c |

Note: Treatments are defined in Table 2. Data is shown as the mean of triplicate measurements. Different letters follow after standard deviation to express significant differences ($p < 0.05$).

## 4. Discussion

Generally, the deep flooding of rice is avoided in production. Long time waterlogging stress leads to leaf senescence and root anoxia, which reduces the ability of rice to absorb nutrients. High temperature is also not conducive to the high yield of rice, which could decrease chlorophyll content of the flag leaf, soluble protein and soluble sugar content of spikelets, increase the contents of malondialdehyde, hydrogen peroxide and free proline in spikelets [24] and reduce the dry matter accumulation of rice [25]. But warming can also increase nitrogen and phosphorus uptake [26,27]. If there is high temperature, there are almost no direct measures to reduce the temperature in the field. Increasing the water depth in the field may slow down heat damage because the existence of these two stresses at the same time may produce beneficial effects. This has not been paid attention to in previous studies.

The aerenchyma areas of H10 and H15 roots were lower than that of H5, indicating that the dual stresses of flooding and high temperature significantly affected the aerenchyma area (Table 1) and delayed the senescence of root parenchyma cells. Because higher temperature enhanced transpiration of rice and promoted the increase of root activity, thus delaying root senescence, programmed cell death would form aerenchyma [28]. However, the formation of aerenchyma is regulated by many factors such as ethylene [29] and nitrogen [30], needing further study.

As an important link between above-ground plant parts and the soil, the root system is involved in the absorption of water and nutrients, the synthesis and transportation of plant hormones and the anchorage of plants [31]. Well-developed rice roots increase biomass and yield in response to different conditions in different cultivars [32]. Root diameter and stele diameter determine the radial transport distance of water and nutrients in roots, and changes in those two parameters affect the absorptive capacity of roots conversely. Fortunately, roots adapt to the environment by changing their shape (such as reducing their diameter) [33]. In this experiment, compared with high temperature treatment and flooding treatments, H10 and H15 decreased the thickness of the outer root, which could improve absorption rates [34]. Furthermore, the root diameter and vessel diameter of the H5 treatment were lower than those of M5, which could reduce the water conductivity of roots. In addition, there was a negative correlation between root diameter and soil temperature (Figure 7), which proved that the increase of air temperature indirectly increased the soil temperature and inhibited root growth. Wu et al. [35] and Majdi and Ohrvik [36] also confirmed our result. The smaller the root diameter, the larger the proportion of root cortex and the smaller root hydraulic conductivity [37]. The vessel diameter of the H15 treatment was lower than that of M5 at the end of stress and recovery of natural growth conditions. The dual stresses of high temperature and waterlogging had a significant effect on vessel diameter (Table 1). As vessel diameter is positively correlated with root hydraulic conductivity [38–41], the dual stresses of high temperature and waterlogging at the booting stage will affect the water conductivity of root system.

High temperature, waterlogging, and their interaction at the booting stage inhibited the development of the rice root system. Qian et al. [42] confirmed that the effect of interaction of

high temperature and waterlogging on cotton yield was greater than that of high temperature and waterlogging alone. However, for rice, the interaction of temperature and waterlogging had less effect on chlorophyll and soluble sugar than temperature or water stress alone [43] and high temperature and waterlogging increased the leaf area and dry matter accumulation of the shoot [44]. This is because rice is a wetland crop, which has a stronger ability to adapt to an anoxic environment than cotton does. High temperature and high humidity conditions enhance the transpiration of rice leaves, which could promote the accumulation of assimilates to leaves [45,46]. Therefore, during the occasional high temperatures after rainstorms, the water depth in rice paddy fields should be maintained at 10~15 cm for five days to not only alleviate the effect of high temperature on rice growth, for example, enhancing the activity of antioxidant enzymes in rice leaves [43], but also to reduce the loss of nitrogen and phosphorus [47].

## 5. Conclusions

(1). Compared with M5 (normal temperature), a combination of high temperature and waterlogging stresses at the booting stage promoted the early formation of aerenchyma in rice roots, and had a certain sustainability but, compared with H5, their interaction lightened the effect of high temperature. High temperature alone (H5) inhibited the expansion of root diameter, stem diameter, thickness of the outer root depth and vessel diameter in rice but waterlogging alone had little effect. As normal growth conditions were restored, the root anatomical parameters of temperature and waterlogging treatments were still inhibited. Since there is relatively little growth of new roots in rice following booting, this root architecture is likely to persist into the grain-fill period, potentially affecting water and nutrient transport and grain yield.

(2). The interaction of high temperature and waterlogging stresses at the booting stage had little effect on rice root diameter, stele diameter and thickness of the outer root depth. However, high temperature × heavy waterlogging (H15) reduced the rice root vessel diameter compared with M5, while it was increased compared with H5, indicating that waterlogging enhanced the ability of rice under high temperature stress.

(3). Temperature and water stress alone at the booting stage affected significantly the root diameter, stele diameter and vessel diameter, whereas their interaction had a significant effect only on vessel diameter, and the interaction of high temperature and waterlogging did not cause more serious damage to rice root structure. According to this, we believe that increasing the depth of water storage after a rainstorm can help reduce high temperature stress in rice.

**Author Contributions:** Conceptualization, B.Z. and X.Z.; methodology, H.L. (Hongfei Lu); software, H.L. (Huizhen Li); formal analysis, Q.N.; data curation, H.L. (Huizhen Li); writing—original draft preparation, B.Z.; writing—review and editing, B.Z.; supervision, H.Q.; project administration, G.T. All authors have read and agreed to the published version of the manuscript.

**Funding:** This research was funded by the National Key R&D Program of China, grant number 2018YFC1508301 and Central Public-interest Scientific Institution Basal Research Fund (No. FIRI202001-04, FIRI2019-05-05).

**Conflicts of Interest:** The authors declare no conflict of interest.

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
