# Peer review of "Effects of Combined High Temperature and Waterlogging Stress at Booting Stage on Root Anatomy of Rice (Oryza sativa L.)"

_water, doi:10.3390/w12092524_

Round 1

Reviewer 1 Report

This is a moderately well written paper based on what appears to be a good data set. However the data as presented are not convincing and the significance of the findings cannot be appreciated without a significant investment in time by the reader. The story is not clear enough for publication and I recommend a major rewrite to present the story more clearly. Firstly I suggest the treatments be labelled as H5 (high temperature 5 cm water depth), H10 and H15 in place of T1, T2 and T3, and M5 (moderate temperature 5cm water depth), M10 and M15 instead of CK, T4 and T5. This will be easier for readers to follow without having to refer to the definition of the treatments. Secondly I suggest that presentation of results start with a summary of the statistically significant effects. Figures 1-4 should be illustrative of effects that should have been proven statistically significant. Table 5 is a good start, but it is unclear which sampling time it refers to. Instead, an additional 4 rows should be added to Tables 1-4 for …

5% Duncan’s Multiple Range

P value of the temperature effect

P value of water effect

P value of the temperature x water interaction

This way a separate analysis is presented for each sampling time.

The Results section is about twice as long as it should be. Only statistically significant effects require description in the text, and generally a table should be described in only 1-2 sentences, highlighting only the features that are followed up in the Discussion. Several sections of the current version of the text describe an interaction between temperature and water level, but there is only one significant interaction shown in Table 5. Few readers have the time to check for letter differences in Tables 1-4, and most will look for a significant overall effect of temperature or water from the analysis of variance before comparing treatment values. According to Table 5 there were no interactions in root diameter, outer root thickness, or stele diameter, with the effects of temperature and water level only having additive effects. The only interaction was in vessel diameter, and the text should make it clear whether this was a positive or negative interaction. Positive is that the combined effects of temperature and water level are greater than their individual effects.

Specific comments

Too many significant figures are usually provided, which makes the document difficult to read. Two significant figures are generally sufficient in the text unless quoting values from the Tables. For example, line 46 should refer to only 44% rather than 44.1%. In the Tables, 3 significant figures should be sufficient, and there is no need for the SE of the 3 replicate values if an overall 5% Duncan’s Multiple Range is provided. For example, the first value in Table 1 should be “448c”.

It is unclear what zero days of stress refers to. Zero days would normally be interpreted as before any stress is imposed. Should this instead be after one day of stress (ie Day 1)? If there are significant treatment effects prior to the imposition of stress, this could be due to poor randomisation. Or that due to random variation alone we would expect one significant difference for every 20 comparisons.

There is text comment about the area of aerenchyma described in Figs 1-4. It would be good if there were a quantitative description, which could be analysed statistically, as for the other parameters. If this is not available for all replicates, an estimate of the area of aerenchyma in Figs 1-4 would enable quantitative description in the text.

Line 45-46 Reword to “…a combination of heat and waterlogging stress occurred 15 times between 1960 and 2010, whereas there were only a further 19 crop failure events that were caused by a single one of these stresses.

Line 49-50 As a reader familiar with rice growing in flooded ricefields, I was surprised by this sentence. Rewrite as “…and instead of the shallow flooding (< 5cm) for which the rice plant is anatomically adapted, the surplus water causes deeper ponding, resulting in yield loss.”

Line 51 “the dual stresses of”

Line 52 Omit “Namely”

Line 55 ”a study”

Line 57 “under these dual stresses from an anatomical perspective”

Line 60-61 “changing homeostatic mechanisms and root architecture [17]. Milder water stress increases root length, .. rate, but the average root diameter decreases”

Line 72-75 "high air temperature leads to high soil temperatures, which in turn affect root water absorption. High root temperature accelerates the aging process, causing the lignified section of the root to extend almost to the tip, leading to a decrease in the absorption area of the root, and the rate at which the root can absorb water and nutrients.”

Line 78 “Substantial research has been undertaken on the effect of a single stress on the water absorption capacity of rice roots.”

Line 80-82 “Anatomical features of the rice root system include root diameter, cross-section of aeration tissue, Stele diameter…cells, all of which affect..”

Line 97 “the tillage layer at the research station”

Section 2.2 Were plants raised outdoors until placed in the growth chambers? This needs to be described.

Line 103 Move harvest details to line 125

Lines 103-109 Most of this should be deleted as it has been covered in the Introduction. That which has not been covered should be moved to the Introduction. The methods section should describe in sequence what was done experimentally.

Line 112 and elsewhere. Change “layer” to “depth”

Line 123-125 There is no need to explain what happened to the plants after exposure to the treatments unless further measurements of these plants are reported in this paper.

Line 144-145 There is no need to include ethics considerations as this is not required for plants.

Line 180-186 “This..[24]” is Discussion. A Results section should only refer to results presented apart from a minimum of interpretation.

Line 223 ”Only the H5 and M5 treatments had some…”

Lines 240-241 “reduced conductivity of the roots”. I don’t follow the logic here. Water conductivity is most likely related to the fifth power of the radius of vessels (Table 4), not the external diameter of the roots. If there is a relationship between the radius of vessels and the external root diameter, this should be presented as a regression analysis.

Lines 312-320 Delete to “…[28].” Instead start with a strong statement about results from this study similar to the next sentence.

Line 324-325 This statement would be worthwhile testing with regression analysis of data from this study rather than only quoting another study.

Line 324 and elsewhere change “treated” to “exposed”

Line 354 Delete “but also…phosphorus” This is a new idea not related to this study, and not necessary here.

Line 361 No data are presented on how the root anatomy changed once normal growth conditions were restored. Better to say “Since there is relatively little growth of new roots in cereals following booting, this root architecture is likely to persist into the grain-fill period, potentially affecting water and nutrient transport and grain yield.”

Reviewer 2 Report

The article is interesting, exploring the impact of combined high temperature and water logging stress on root anatomy of rice. 

The authors can include a short discussion in Section 1 explaining the benefit of studying the combined effect of high temperature and water logging rather than studying the two stresses separately, as done in previous research (i.e. is there interaction between high temperature and water logging? What is the problem if we ignore the interaction and focus on the two stresses separately and independent from each other?). 

Correct the spelling of "According" in line 75.

The information discussed in lines 106-110 is very important to show the importance of this research. If this information were to be discussed together with discussions about the expectations about increased temperature and rainfall in section 1, it would have provided a strong argument showing the need for and contribution of this research.

The discussion of the conclusions can be improved. Rather than summarising the results, draw some strong conclusions. I.e. answer the question "So what?". You have found the impact of high temperature, water logging, and the interaction between high temperature and water logging on rice roots, so what? I believe that, if Section 1 is extended to emphasise the need for studying the combined effect rather than the individual effects, the authors will be guided to draw stronger conclusions to show the contribution of this research to new knowledge. 

Reviewer 3 Report

Dear Authors

The aim of the paper titled "Effects of Combined High Temperature and Waterlogging Stress at Booting Stage on Root Anatomy of Rice (Oryza sativa L.)" is  analyze "the effects of high temperature and waterlogging on the growth and development of rice roots by comparing the root diameter, stele diameter, and the number and diameter of vessels between plants exposed to different temperatures and waterlogging regimes, and examined the interaction of high temperature and waterlogging on the absorption of water and nutrients by rice roots".  This  statement was indicated in 83-87 lines, but in abstract is indicated only that paper concerns about "studying the changes in anatomical structure of the rice root system in relation to high temperature and waterlogging stress in rice plants", lines 11-13 and it is with line with title. Only that last sentence is true because in the manuscript is no one result on absorption nutrition.

No design method of experiment towards of evaluation of absorption of nutrition in relation to changes in anatomical roots. No one experimental was made on this aspect , so authors are not authorized to draw conclusions on influence of roots anatomical changes on absorption nutrition, which are put in line 353-354. 

In 2.2. Design of experimental is also not clear explained, it is surprised that authors indicated that regim for development of roots is 32-35 degree C and 5 days with flooding depth 10 cm and more (according to references 19 and 22), lines 107-109, and next six treatments are planed with range 30-34 degree C and 35-38 degreeC. According o this data data range of T4-T6 should be 30-32 degree.  

Also is strange that CK is only for range of temperatures 30-34 degree, for the second range of temperatures 35-38 C degree should be fixed second CK. 

The language is correct, but sometimes the authors write too laconically, which forces the reader to guess, they use mental acronyms, especially in discussions. 

The discussion shows the percentages that are not mentioned in the results chapter. Of course, you can guess that these are the values calculated from the data contained in the tables, but it should be precisely indicated together with the reference to a specific table.

The discussion is laconic and does not bring any new recommendations apart from those contained in literature 19 and 22. The abstract also does not contain any specific information constituting a new conclusion based on the conducted research.

The studies should be planned in a different temperature regime, with two controls and taking into account the analysis of the uptake of e.g. phosphorus and nitrogen (which the authors mention in line 354).

The scope of the literature is limited to rice only, it can of course be understood because of the issue to which the manuscript relates, some of the literature comes from earlier years (this is obviously not a serious objection), but the discussion would be more interesting if other plant species were also discussed in relation to the experimental conditions of the conducted research.

Round 2

Reviewer 1 Report

Comments on article “Effects of combined high temperature and waterlogging…”

This is a significant advance on the version initially submitted, but the Results and Discussion sections are still difficult to read and need extensive rewording and reduction in length that is beyond the role of a reviewer. The Introduction and Methods read well. I suggest the authors look at one more model papers (such as reference 7) and follow their styles for the Results and Discussion sections.

The overall statistical analysis of the effects of temperature vs water depth should be presented first in the Results section (current Table 4), and only effects that are significant in this Table need be presented in the text. Results should be discussed relative to the low stress control (M5), eg a percentage higher or lower. Treatments should be in the order M5 (low stress control), M10, M15, H5, H10, H15. Currently there is too much emphasis on differences between individual treatments. The Results section needs to be fully rewritten and greatly reduced following styles in model papers.

The manuscript often uses the term “waterlogging” incorrectly. All treatments are waterlogged, what was tested was differences in the depth of flooding.

Minor points

Abstract

Line 15 Could be reworded to…”To examine interactions between temperature and waterlogging stress, an experiment was set up in a growth chamber consisting on 2 temperatures (moderate, 30-34C and high, 35-38C) x 3 depths of flooding (0-5 cm, 10 cm and 15 cm). Treatments commenced at booting and lasted 5 days, after which all treatments were returned to a 0-5 cm flooding depth and the same temperature regime. Observations were made immediately after cessation of treatments, then after 5, 10 and 20 days to test the effect of treatments on subsequent root anatomical development. The low-stress control (0-5 cm medium temperature) showed no change with time in aerenchyma area, thickness of the outer root, Stele diameter, and the number nor diameter of xylem vessels.” [Then describe the effects of higher temperatures and deeper flooding relative to this low-stress control.]

Note that there is no need to introduce the treatment designations (such as M5, M10) in the Abstract unless referred to 3 or more times.

Line 48 “accounting for 44%”

Line 55-58 “…in grain yield. Root growth is a dynamic process, and architecture…” (delete sentence that says much the same thing as the following sentence)

Line 59 “on the microstructure”

Line 64 “Shallow flooding increases”

Line 73 “High air temperature”

Line 77 What is “total water cross-section”? Is this the cross-section of fresh roots?

Line 80 Remove and start at “Substantial”

Line 85-86 “[21]. Both high temperature and deep flooding have a negative effect on rice”

Line 92 “the dual stresses of”

Line 119-120 “Experimental treatments of high temperature and flooding depth were imposed”

Line 125-131 …flooding. Temperature treatments were moderate (30-34C) and high (35-38C), while flooding treatments were 0-5 cm, 10 cm and 15 cm. Treatments were designated M5, M10 and M15 respectively for the moderate temperatures and H5, H10 and H15 for the high temperatures. Temperature treatments were imposed in a climate chamber.

Fig 1 Lines should be labelled “Daily maximum temperature” and “Daily minimum temperature”

Line 189 Waterlogging at the booting stage

Line 191 “connected. The aerenchyma”

Line 197 “same flooding depth, high temperature significantly increased the”

Line 308 “the deep flooding of rice”

Line 336 Equation is not needed as it is in the Figure

Line 339 Does a thicker root cortex hinder the lateral movement of water movement into the roots, or the longitudinal movement of water along the roots? Without consulting the references I would expect the longitudinal movement to be most related to the vessel diameter.

Fig 5 should be in the Results

Line 394 Should “litter” be changed to “little” ?

Reviewer 3 Report

Dear authors, 
The article, after taking into account the corrections and suggestions of the reviewers, is properly edited and discussed. It clearly presents the assumptions of the experiment and presents the results without speculating in the conclusions. There is a definite improvement, I have no further comments.

Kind regards,

Author Response

Dear reviewer,
Thank you for reviewing our manuscript.

Kind regards

Bo Zhen